# Nuances in intensity deviant asymmetric responses as a biomarker for tinnitus

**Ekaterina A. Yukhnovich**[ID][1]*, **Kai Alter**[1,2], **William Sedley**[1]

**1** Translational and Clinical Research Institute, Newcastle University Medical School, Newcastle University, Newcastle upon Tyne, United Kingdom, **2** Faculty of Modern and Medieval Languages and Linguistics and the Languages Sciences Interdisciplinary Research Centre, University of Cambridge, Cambridge, United Kingdom

* e.yukhnovich1@newcastle.ac.uk

## Abstract

We attempted to replicate a potential tinnitus biomarker in humans based on the Sensory Precision Integrative Model of Tinnitus called the Intensity Mismatch Asymmetry. A few advances on the design were also included, including tighter matching of participants for gender, and a control stimulus frequency of 1 kHz to investigate whether any differences between control and tinnitus groups are specific to the tinnitus frequency or domain-general. The expectation was that there would be asymmetry in the MMN responses between tinnitus and control groups at the tinnitus frequency, but not at the control frequency, where the tinnitus group would have larger, more negative responses to upward deviants than downward deviants, and the control group would have the opposite pattern or lack of a deviant direction effect. However, no significant group differences were found. There was a striking difference in response amplitude to control frequency stimuli compared to tinnitus frequency stimuli, which could be an intrinsic quality of responses to these frequencies or could reflect high frequency hearing loss in the sample. Additionally, the upward deviants elicited stronger MMN responses in both groups at tinnitus frequency, but not at the control frequency. Factors contributing to these discrepant results at the tinnitus frequency could include hyperacusis, attention, and wider contextual effects of other frequencies used in the experiment (i.e. the control frequency in other blocks).

## 1. Introduction

Subjective tinnitus is a persistent sound heard by an individual without an environmental source, which may appear as pure tone, ringing, hissing, whistling, static, or cicada-like sounds [1, 2]. Due to the variety of potential causes of subjective tinnitus, classification of this condition has been difficult [3]. Heterogeneity of tinnitus mechanisms, along with the existence of specific categories, or even a continuous spectrum of tinnitus, remains controversial [4–6]. It is possible that many potential biomarkers of tinnitus and treatment options are applicable only to certain individuals [7–10]. As such, there is a need for a biomarker that would indicate the presence of tinnitus across all the potential tinnitus subgroups or dimensions. This would need to relate to mechanisms forming part of a 'final common pathway' for tinnitus,

**Data Availability Statement:** The data can be accessed in the Newcastle University data depository: 10.25405/data.ncl.23662047.

**Funding:** This work was supported by the Royal National Institute for Deaf People and the Masonic

Charitable Foundation, United Kingdom (EAY, WS). The funders had no role in study design, data collection and analysis, decision to publish, or preparation of the manuscript.

**Competing interests:** The authors have declared that no competing interests exist.

**Abbreviations:** DD, Downward Deviant; EEG, Electroencephalogram; HQ, Hyperacusis Questionnaire; IMA, Intensity Mismatch Asymmetry; MMN, Mismatch Negativity; PTA, Pure Tone Audiogram; THI, Tinnitus Handicap Index; UD, Upward Deviant; ULL, Uncomfortable Loudness Level.

irrespective of specific contributory mechanisms. Such a biomarker might help to better understand tinnitus mechanisms and allow treatment studies to determine the effectiveness of their treatment across different tinnitus groups [11] It may also be possible to translate this biomarker into animal models, which would then allow to differentiate between hearing loss, hyperacusis and tinnitus more accurately than presently possible, and thus improve tinnitus research in animals [2].

A biomarker based on the Sensory Precision Model of Tinnitus in humans might contribute towards this goal of a single invariant biomarker, as it encompasses the variety of causes and contributors, and specifies a single mechanism through which they interact to cause tinnitus [12]. The Sensory Precision Integrative Model of Tinnitus is based on predictive coding [12, 13]. In this model, spontaneous activity is always present in the auditory system; however, its signal tends to be incoherent, and weaker than true sensory input [13]. According to this model, such a 'tinnitus precursor' generates spontaneous prediction errors. As the tinnitus precursor is not correlated to internal or external events, and is not behaviourally relevant, it has low 'precision'. As such, when compared against top-down prior predictions, or competing bottom-up inputs, the tinnitus precursor is usually explained away as noise. However, a tinnitus precursor may become sufficiently intense, or be given too much precision, leading to a false rejection of the default null hypothesis of 'silence'. Accepting the tinnitus precursor as a true 'signal' thereby reduces the prediction error it generates. Once the tinnitus precursor is perceived, eventually, the repeated rejection of silence as the baseline perceptual state may create a new default prediction of tinnitus. Associative plasticity and other forms of learning at higher levels (e.g., parahippocampally mediated memory) might allow the perception of tinnitus to continue even after the factors that increased the intensity, and/or precision, of the precursor are removed. Although these changes in the tinnitus precursor (which may manifest as altered spontaneous neural activity in the forms of firing rates, neural synchrony, large-scale oscillations, and metabolic or blood flow changes), may be temporary, and therefore might not be detectable in the long-term with conventional neuroimaging methods, the skewed default predictions may be detectable as altered prediction error responses to specific auditory stimuli around the tinnitus frequency. One such commonly used measure of prediction violation is the mismatch negativity (MMN). MMN is an evoked potential that indicates auditory change detection based on recent auditory context, irrespective of whether attention is aimed away or towards the stimulus [14–16].

Motivated by the Sensory Precision Integrative Model, Sedley et al (2019) used a roving intensity paradigm to elicit MMN in people with chronic tinnitus compared to age and hearing matched controls [1]. The roving paradigm is a type of oddball paradigm, but with two types of standard stimuli where deviants are defined as pseudo-random transitions between one standard type and the other. The high intensity (loud) standard was interrupted by a quieter (downward) deviant, while a low intensity (quiet) standard was interrupted by a louder (upward) deviant. The findings indicated that, in response to sounds of a frequency that was similar to their tinnitus, participants with tinnitus had larger MMN responses to upward deviants, but smaller MMN responses to downward deviants, compared to the control group. No relationship was seen between MMN, and Tinnitus Handicap Inventory (THI) [17] or visual analogue scale of tinnitus loudness score (subjective measure of tinnitus loudness); this finding was termed 'Intensity Mismatch Asymmetry' (IMA). The hypothesis was that downward deviant stimuli sounded more similar in intensity to the default prediction of tinnitus intensity, so people with tinnitus showed a reduced response to this change as it was a more expected sound. However, the upward deviant was further from the default prediction, thus making the MMN response larger. However, because only stimuli at or close to the tinnitus frequency (or frequency band) were tested, it is not known whether this asymmetry of intensity mismatch

responses in people with tinnitus compared to controls is specific to the tinnitus frequency or generalised across frequencies.

The current study attempted to replicate the findings of the original roving intensity paradigm, with an addition of a control frequency that is far from the tinnitus frequency, to see whether differences seen between tinnitus and control subjects were: 1) replicable, and 2) frequency-specific.

## 2. Materials & methods

### 2.1 Participants

Volunteers with tinnitus (n = 14) were recruited from affiliated volunteer lists at Newcastle University. The sample size was small largely because this study was seeking the kind of large and invariant effects indicated in the previous study [1], and whether those findings could be help up in another group. To be included, participants needed to be over 18 years of age, with chronic tinnitus for over 6 months that did not have a physical source and was not due to Meniere's disease, who could make an informed choice about volunteering. Exclusion criteria included using ongoing sedating or nerve-acting medications, and mental health conditions severe enough to interfere with everyday life activities. Non-tinnitus participants were recruited using the same mailing lists, and individually matched to tinnitus participants, based on an approximate match of their overall audiometric profiles, with particular attention to the vicinities of 1 kHz and the tinnitus frequency. It was also ensured that there were no significant group differences between tinnitus and control groups in age or sex.

Recruitment and data collection occurred between November 2019 and June 2021. Participant data was anonymised after data collection with the use of a participant number. Approval was given by the Newcastle University Research ethics committee, and all participants gave written informed consent according to the Declaration of Helsinki (reference number 5619/2020).

### 2.2 Psychophysical assessment

All research activity took place within the Auditory Cognition Lab, Newcastle University. Subjects completed a short demographic questionnaire, with additional questions about any health conditions or medications, and the Hyperacusis Questionnaire (HQ) [18]. Participants with tinnitus also completed THI. All participants underwent pure tone audiometry at 0.25, 0.5, 1, 2, 4, 6, and 8 kHz.

Tinnitus participants underwent two computerised tasks, performed under supervision. In the first, they performed 5 rounds of tinnitus matching of a random sound generated by Matlab (The MathWorks). In each round of matching, they tuned an ongoing synthetic band pass noise stimulus with random starting parameters in real time in frequency, bandwidth, intensity, and laterality balance. At the narrowest bandwidth, the stimulus became a pure tone. In cases of bilateral tinnitus, the participants heard the sound in both ears, and could adjust the ear balance. The intensity was based on an inverse Fourier transform of a Hanning spectrum noise, with peak amplitude at the centre frequency equal to 1. It was not a specific dB value, and it was always a relatively quiet stimulus which subjects needed to increase in intensity for the match.

Participants could discard any matches they felt were not close to their tinnitus. The average of the remaining matches was taken to be used in the second task as an indicative tinnitus match, to form the starting point for individual experimental stimulus determination. In this next task, tinnitus participants were presented with pure tones whose frequency was determined based on their average tinnitus match. Then, the experimental stimulus frequency for

this experiment was determined using the same process as the edge frequency calculation (i.e., the lower spectral edge of the tinnitus match) in the original experiment, because the IMA effect was stronger when participants were presented with the edge frequency rather than the centre frequency of their tinnitus [1]. To achieve this edge frequency, participants were asked to ensure that it was slightly below the lower spectral edge of their tinnitus (i.e., that they could discern the tones and their tinnitus as two distinct non-overlapping sounds). They were able to adjust the frequency of these tones, if needed, until they were satisfied that they had found the edge frequency of their tinnitus. They were then asked to adjust this sound to a comfortable but loud volume. They then adjusted the intensity of 1 kHz pure tones until they matched the subjective loudness of their tinnitus edge frequency tones. These two stimulus intensities (one for tinnitus edge and one for control frequency) were designated the 'high' stimulus intensities for the main experiment, with 'low' intensities set 6 dB lower than this. A final check was performed to ensure that subjects could hear both 'low' intensity stimuli and distinguish them as subjectively quieter than the 'high' intensities. In cases where these criteria were not both met, subjects could increase or decrease the intensity of the 'low' intensity stimuli, to ensure that they were both audible and differentiable from the 'high' intensity stimuli. In other cases, the 6 dB intensity difference was maintained. Control participants were allocated the same experimental frequency as their matched tinnitus subject and had full control over stimulus intensities as for the tinnitus subjects.

## 2.3 Experimental design

EEG was recorded in a soundproof room, using a 64 channel Active two system (Biosemi). No ocular channel was used as the standard EEG channels were sufficient for removing ocular artifacts. Participants watched a silent subtitled movie of their choice, while the stimuli were played to them through headphones. Electrode offset was kept at manufacturer-recommended limits of +/- 10 mV.

The experimental design closely followed the paradigm used in the original study [1], with the additional inclusion of a 1 kHz tone as a control condition. The roving paradigm employed in this study is a type of oddball paradigm, but with two types of standard stimuli, and where deviants are defined as pseudo-random transitions between one standard type and the other every 4 to 8 stimuli. Stimuli were 300 ms tones, with 10 ms onset/offset ramps, followed by 300 ms inter-stimulus intervals. The tones were presented isochronously to the ear(s) that the tinnitus participant indicated as the tinnitus ear(s), or the same ear(s) for their matched control. For example, if the participant with tinnitus only had tinnitus in their right ear, the matched control would also only hear the tones in their right ear. The high intensity (loud) standard was interrupted by a quieter (downward) deviant, while a low intensity (quiet) standard was interrupted by a louder (upward) deviant. There was also a duration deviant condition, in which a duration deviant tone of 150 ms was followed by a 450 ms gap every 1 out of 10 stimuli. The purpose of duration deviants was to assess for the presence or absence of more general auditory mismatch detection differences associated with tinnitus.

## 2.4 EEG data processing

Data analysis was performed in Matlab, using the EEGLAB toolbox [19]. Data were down sampled to 256 Hz from the original 1024 Hz, and re-referenced to combined P9/P10 channels, approximating to linked mastoids. Data were then filtered using a high-pass cut-off of 0.3 Hz and a low-pass cut-off of 25 Hz. Bad channels were removed using 0.8 as the minimum acceptable correlation with nearby channels. The removed channels were then

reconstructed through interpolation. Data were then epoched between -0.1 and 0.5 s peristimulus time. Denoising Source Separation [20] was used as to remove artefacts. The first four components were retained for all subjects, based on prior inspection of all subjects' data in order to achieve an optimal balance between preserving signal and eliminating noise. The data were then put through EEGLAB automatic artefact rejection using probability of 5 and kurtosis of 8. The epochs were baseline corrected to -100-0 ms peristimulus time.

## 2.5 Statistical analysis

Statistical analysis was performed using MATLAB. To compare the evoked responses in participants with tinnitus and controls, a three-way ANOVA was used, with subject group, frequency, and intensity used as factors of interest, and including interaction terms. Additionally, two-way ANOVAs were used to look at each frequency separately due to differences between the two frequencies potentially overshadowing any differences between subjects and intensities. Additionally, two-way ANOVAs were carried out to look at the duration deviants.

## 3. Results

### 3.1 Demographic information

Both volunteer samples each comprised of 14 right-handed participants (overall n = 28), matched based on age, gender and hearing measured with pure-tone audiometry. The age and hyperacusis information for all participants can be found in Table 1. In the tinnitus group, 5 people indicated presence of hyperacusis according to the HQ, when following the more stringent but potentially more sensitive thresholds of >16 [21].

Tinnitus matches, as well as general information about the condition in the tinnitus group can be found in Table 2. According to the THI, tinnitus was causing a slight problem to four participants (<16 THI score), a mild problem to seven participants (18–36), a moderate problem to two participants (38–56) and a severe problem to one participant (58–76). Tinnitus duration ranged from 4 to 60 years. The duration was skewed to the left, so the mean (21) was lesser than could be expected in a normal distribution. Seven participants described hearing their tinnitus centrally/equally between the ears; two participants had tinnitus more in the right ear than the left; one had it more in the left ear than the right; two had it mainly in the left ear; finally, two participants had it entirely in the left ear. Eleven participants described their tinnitus as whistling/ringing/pure tone. One person indicated having both ringing and hissing sounds. One person stated they had hissing/static. The last person described their tinnitus as 'staccato sounds'. Half the sample said their tinnitus fluctuated over days/weeks. Nine participants indicated that their tinnitus became worse during/after being in loud environments. The

**Table 1. Age and hyperacusis results in both groups, which were non-normally distributed.**

| Demographics | Tinnitus Group Mean (Standard Deviation) | Control Group Mean (Standard Deviation) | Comparison (Mann-Whitney U) |
|---|---|---|---|
| Age | 52.64 (21.24) | 50.57 (StdD 18.34) | p = 0.734 |
| | Range: 20–80 | Range: 22–70 | |
| Hyperacusis | 16.57 (7.83) | 6.64 (StdD 4.99) | p = 0.001* |

Significant differences between the groups highlighted with *.

**Table 2. Tinnitus questionnaire findings.**

| Measure | Mean (Standard Deviation) |
|---|---|
| THI | 27.29 (16.69)* |
| Duration (years) | 21 (19.16) * |
| Loudness on average (0–10) | 5.14 (1.83) |
| Bothersomeness (0–10) | 4.57 (2.65) |
| Loudness on the day (0–10) | 5.57 (1.56) |
| Aware of tinnitus during the day (%) | 51.79 (33.43) |
| Average tinnitus edge frequency (Hz) | 5075.91 (1747.52) |

Loudness, and bothersomeness were measured using a scale 0 to 10, with 0 being not at all and 10 being the most possible. Tinnitus awareness was measured out of 100%, 100% representing the entire day.
* Indicates non-normally distributed data, according to Shapiro-Wilk's test.

average tinnitus edge frequency in Table 2 was the frequency chosen by the participant to represent the lowest frequency of their tinnitus.

There were no significant differences in hearing thresholds between groups at any of the frequencies measured (p = 0.851; p = 0.138; p = 0.753; p = 0.627; p = 0. 216, p = 0.085; p = 0.087, for frequencies in ascending order) (Fig 1). Thresholds at each tinnitus frequency linearly interpolated for each individual were also not significantly different between groups (p = 0.271).

In the tinnitus group, 5 participants left the relative difference in intensity between loud and quiet stimuli at -6 dB, with the rest adjusting the difference in intensity. In the control sample, 8 participants left the difference in the volume of the stimuli as set by their counterpart with tinnitus; 4 control participants also left the difference at -6 dB. Nonetheless, the mean dB difference between the loud and quiet stimuli was not significant between the tinnitus and

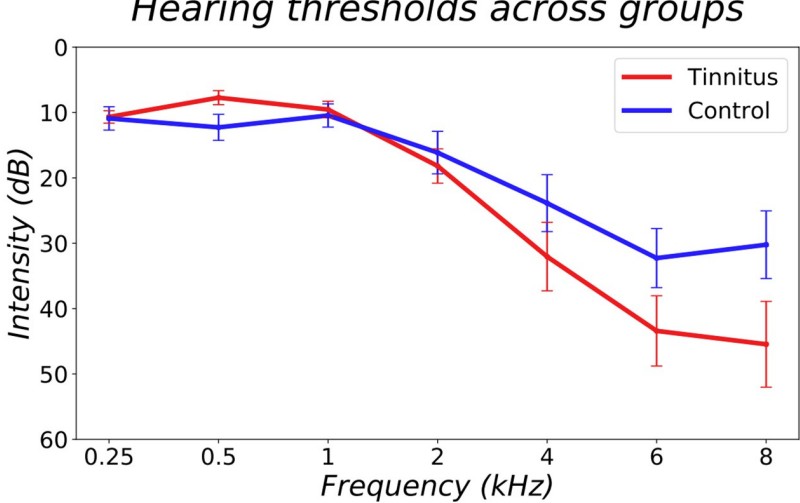

**Fig 1. Mean hearing threshold of the tinnitus (red) and control (blue) groups at 0.25,0.5,1,2,4,6 and 8 kHz.** Tested using a pure tone audiometer. Data for each frequency, except 2 kHz, was not normally distributed.

**Table 3. Audiometry table.**

|  | Tinnitus Group Mean | Control Group Mean |
|---|---|---|
| 1 kHz (dB HL) | 9.55 (5.96) | 10.45 (8.30) |
| 1 kHz (dB SPL) | 56.01 (10.06) | 63.56 (9.93) |
| 1 kHz sensation level (dB) | 46.46 (10.03) | 53.11 (12.16) |
| Edge frequency (dB HL) | 34.35 (21.80) | 27.33 (19.88) |
| Edge frequency (dB SPL) | 68.31 (18.36) | 75.48 (17.32) |
| Edge frequency sensation level (dB SL) | 33.96 (15.45) | 48.15 (20.02) |

The edge frequency audiometric threshold was calculated linearly for each participant. dBHL represents the mean PTA thresholds for each group; dBSPL represents the mean intensity at which the stimuli were set to play through the headphones by the participants in each group; dB SL represents the mean perceived intensity by the participants (dBSPL-dBHL).

control groups (Man U test, p = 0.376). To investigate any systematic differences in the stimulus intensities between the two groups, sensation levels were calculated, i.e. the difference between hearing threshold (dB HL) and stimulus intensity (db SPL) as measured by a sound level meter (Table 3).

In the right ear, there was no difference in the sensation levels between the tinnitus and control groups at 1 kHz (mean = 47.95 and 53.00, respectively; t(34) = 0.038, p = 0.113), but there was a significant difference at the tinnitus edge frequency (mean = 31.40 and 52.85, respectively; t(34) = 1.74, p <0.001).

In the left ear, there were singificant differences in the sensation levels between tinnitus and control groups at both 1 kHz (mean = 45.43 and 53.18, respectively; t(50) = -2.26, p = 0.029) and tinnitus edge frequency (mean = 35.74 and 44.90, respectively; Mann-Whitney U, p = 0.011). Overall, the tinnitus group received stimuli with lower sensation levels than the control group. These findings were not related to HQ scores.

## 3.2 Spatiotemporal organisation of stimulus response

Grand average ERP data for channel FCz across all stimulus conditions and subjects (Fig 2a) was used to determine timeframes for quantifying P50, N100 and MMN responses (Table 4), based on visual inspection. Difference waveforms were calculated by subtracting standard responses from their equivalent deviant conditions (b,c,d,e).

## 3.3 Early evoked potentials (P50, N100) are not affected by tinnitus

**3.3.1 Standard and deviant stimuli are affected by frequency.**   Fig 3 shows P50 responses to standard stimuli (a, b). A three-way ANOVA (subject group, stimulus frequency, stimulus intensity) showed a main effect of stimulus frequency in P50 responses to these stimuli (p = 0.0003). No other significant effects were identified.

N100 findings were similar to P50 findings in showing similar responses in both tinnitus and control groups (Fig 4a and 4b). A main effect of frequency was found in a three-way ANOVA (p = 0.0005).

**3.3.2 Difference waveform.**   The difference waveform between standard and deviant stimuli was also investigated in the P50 and N100 timeframes to ensure that any differences seen in the MMN timeframe were not dependent on differences in earlier stages of processing carried forward. No significant effects were found within the P50 timeframe. The N100 difference

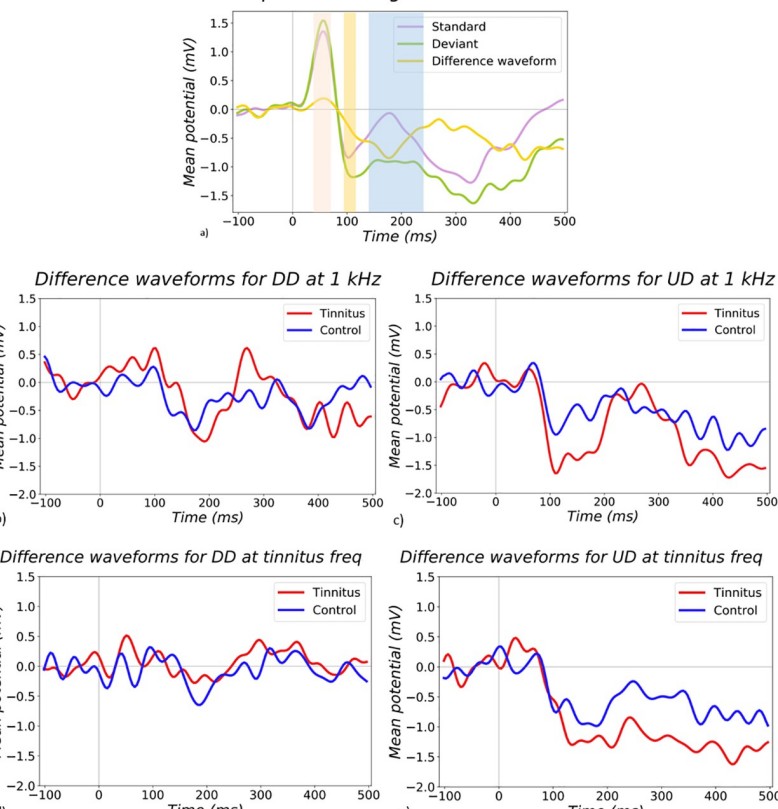

**Fig 2. Timelines of responses.** a) The plot shows the averaged standard, deviant, and difference waveforms to all intensity conditions (Upward Deviant (UD), Downward Deviant (DD), Standard Quiet, Standard Loud). The mean potential in microvolts is on the y-axis, which is plotted against the timeline, where 0 ms is the stimulus onset (shown as a vertical grey line). These responses are averaged across tinnitus and control groups to avoid any bias towards timelines in either group. Graphs b),c),d) and e) show difference waveforms separated by subject group and stimulus condition. Tinnitus group is shown in red and control group is shown in blue. The MMN responses at the control frequency (1 kHz) to DD are shown on graph b) and to UD on graph c). The MMN responses at the tinnitus frequency to DD are shown on graph d) and to UD on graph e).

waveform showed a main effect of deviant direction (p<0.0001) (Fig 5). A stronger negative response was seen to the upward deviant.

Additionally, a three-way ANOVA (subject group, stimulus frequency, stimulus intensity) indicated a main effect of directionality in a late negative potential (280–500 ms; Fig 2), which was greater in upward deviants than downward deviants (p = 0.0002) but was not influenced by tinnitus status.

**Table 4.  Timeframes for analysis of P50, N100 and MMN ERPs based on Fig 2a.**

| ERP | Timeframe (ms) |
| --- | --- |
| P50 | 40–75 |
| N100 | 93–114 |
| MMN | 140–243 |

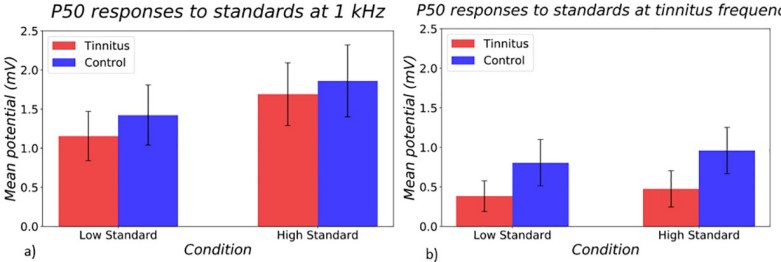

**Fig 3. P50 responses in tinnitus (red) and control (blue) groups to standard stimuli at the a) control frequency and at the b) tinnitus frequency.** On the left hand side of both bar graphs, responses to the quiet standard is shown, and on the right, responses to the loud standard is shown.

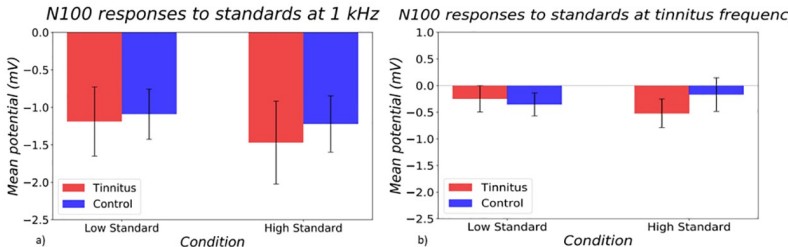

**Fig 4. N100 responses in tinnitus (red) and control (blue) groups to standard stimuli at the a) control frequency and at the b) tinnitus frequency.** On the left hand side of both bar graphs, responses to the quiet standard is shown, and on the right, responses to the loud standard is shown.

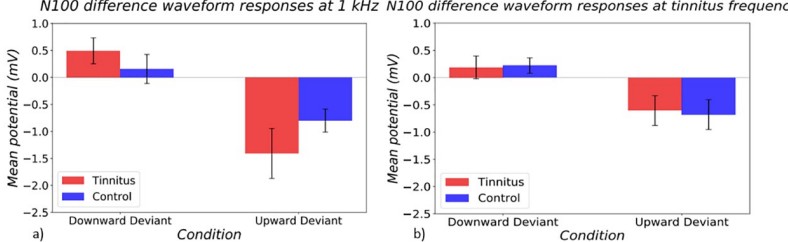

**Fig 5. Difference waveforms between deviant and standard responses at N100 timeframe at a) control frequency and b) tinnitus frequency in tinnitus (red) and control (blue) groups.** On the left side of both graphs, responses to the DD condition are shown, and on the right side, responses to the UD condition are shown.

## 3.4 Direction of the deviant affected responses in the MMN timeframe

As we included duration deviants in this paradigm, a similar analysis was carried out for these as the intensity deviants.

**3.4.1 Standard stimulus responses.**   Responses in the MMN timeframe to standard stimuli (Fig 6a and 6b) in a three-way ANOVA (subject group x frequency x intensity) showed a non-significant trend towards a main effect of larger responses in tinnitus subjects (p = 0.061).

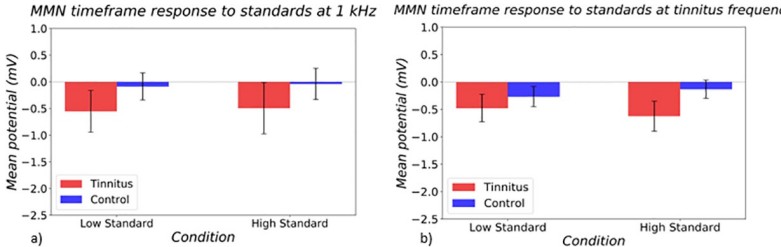

**Fig 6. MMN timeframe responses to standard stimuli at a) control frequency and b) tinnitus frequency in tinnitus (red) and control (blue) groups.** On the left hand side of both bar graphs, responses to the quiet standard is shown, and on the right, responses to the loud standard is shown.

**3.4.2 MMN difference waveforms.** The main response of interest was the MMN difference waveform between deviants and standards, with the expectation being to see a group x direction interaction (if the previously observed effect generalised across frequencies), or a group x frequency x direction interaction (if the effect did not generalise). A three-way ANOVA (subject group, stimulus frequency, stimulus intensity) showed a main effect of deviant direction in the MMN responses of the participants ($p = 0.049$) (Fig 7), with larger responses to upward intensity deviants. Unlike the original study, we did not observe a significant difference in asymmetry in deviant direction responses between tinnitus and control groups (group x direction interaction $p = 0.239$; group x direction x frequency interaction $p = 0.752$). The pattern seen in the tinnitus group was similar to the pattern seen in the original study, in showing larger MMN responses to upward than downward intensity deviants, though this deviant direction effect did not quite reach significance even when analysed in tinnitus group only ($p = 0.058$). However, the results from the control group in the present study were different to those from the original study; in the present study, responses from control subjects followed a similar pattern to the tinnitus group at the tinnitus frequency (Fig 7b). The similar findings in the tinnitus and control groups were reflected in a two-way ANOVA (subject group and deviant direction as factors) showing a main effect of deviant direction at the tinnitus frequency ($p = 0.015$), but not at the control frequency ($p = 0.811$). The group x direction interaction was not close to significance in either of these analyses ($p = 0.305$ for control frequency, and $p = 0.534$ for tinnitus edge frequency). In summary, MMN responses at the tinnitus edge frequency in the present study were similar to previous study (Sedley et al, 2019) for

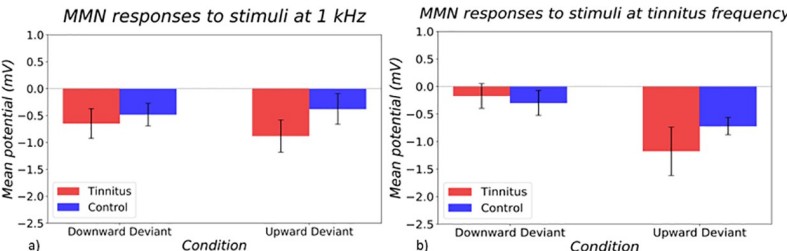

**Fig 7. Difference waveforms between deviant and standard responses at a) control frequency and b) tinnitus frequency in tinnitus (red) and control (blue) groups.** On the left side of both graphs, responses to the DD condition are shown, and on the right side, responses to the UD condition are shown.

the tinnitus group, but different in the control group, who in this study appeared more similar to the tinnitus group. Conversely, the control stimulus frequency showed a very different intensity MMN pattern, with no clear directional asymmetry.

## 3.5 Duration deviants

Similarly to the intensity MMN, timeframes for the duration MMN was chosen based on inspection of grand average ERP data across all stimulus conditions (Fig 8a). The MMN timeframe chosen was 220–340 ms. Standard responses appeared similar across groups and conditions upon visual inspection (Fig 8b), whereas difference responses appeared somewhat larger in the tinnitus group at the control frequency compared to the control group (Fig 8c). This difference, however, was not significantly different based on an ANOVA (subject x frequency) (p = 0.099). However, when looking specifically at deviant responses rather than difference waveforms, an ANOVA (subject x frequency) showed a main effect of frequency (p = 0.009) (Fig 8d and 8e).

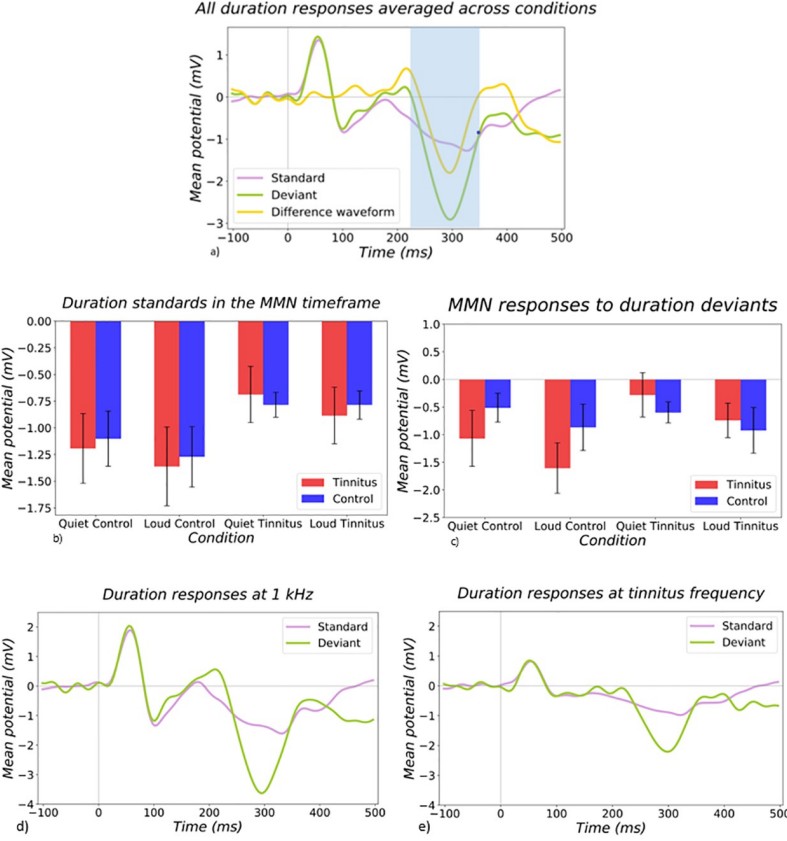

**Fig 8. Responses to standard and deviant duration stimuli.** a) The plot shows the averaged standard, deviant, and difference waveforms to all duration conditions (shorter duration, standard duration). The mean potential in microvolts is on the y-axis, which is plotted against the timeline, where 0 ms is the stimulus onset. Graph b) shows responses to the standard duration stimuli duration each of the intensity conditions (control = control frequency, tinnitus = tinnitus frequency). Tinnitus group is shown in red and control group is shown in blue. Graph c) shows the MMN responses to duration deviant stimuli during each of the intensity conditions. To look more closely at the pure deviant responses at each of the frequencies, graph d) shows responses to standard (purple) and deviant (green) stimuli at the control frequency and e) shows responses to standard and deviant stimuli at the tinnitus frequency.

## 4. Discussion

### 4.1 Differences between current and original studies are likely due to the control sample

This study was carried out to replicate previous findings [1]. The expectation was that there would be asymmetry in the MMN responses between tinnitus and control groups at the tinnitus frequency, but not at the control frequency, where the tinnitus group would have larger, more negative responses to upward deviants than downward deviants, and the control group would have the opposite pattern or lack of a deviant direction effect. There were no differences between the groups for the control frequency, as was anticipated, with similarly sized responses to upward and downward deviants. The pattern seen in the tinnitus group was similar to the pattern seen in the original study. However, the control group in the present study followed a similar pattern to the tinnitus group at the tinnitus frequency. A number of technical considerations could be involved in this finding, such as the smaller sample size and individual differences in the subjects (e.g. overall higher hearing loss levels at higher frequencies in this experiment), as well as inter-researcher differences in the implementation of methods. The current study also tightly controlled for gender. There were stimulus properties differences between the two groups, with the tinnitus group receiving perceptually quieter sounds. Nevertheless, this was unexpected and may require further investigation.

### 4.2 Upward intensity deviants may elicit stronger negative ERP components than downward deviants

In the current study, the upward deviants caused a stronger N100 and a late negative response, when compared to the downward deviants in both groups and frequencies. This is in accordance with recent unpublished data from our group, where N100 was also significantly increased for upward but not the downward deviants, compared to their respective standards. Previous research showed similar patterns to sound intensity changes [22–24]. These researchers used 1 kHz tones that were either 50/60 dB or 80 dB, which is a much larger difference in intensities than used in the studies in our lab, however, the overall paradigm seems fairly similar so that comparisons could be drawn. Additionally, a stronger late negative potential has been previously shown in relation to upward deviants, and to metrically accented sounds along with the N100 [22, 25]. Therefore, stronger negative ERP components might be expected in response to the louder deviant sounds in such paradigms.

Additionally, there is a striking finding of stronger early components of responses to the standard stimuli in the control frequency compared to the tinnitus frequency. This may be an intrinsic property of the brain response, or it could reflect the hearing loss both participant groups exhibited at higher frequencies.

### 4.3 Stimulus frequency differences may influence intensity mismatch asymmetry (IMA)

As early ERP components were significantly affected by the frequency of the stimuli, it is possible that incorporating a control frequency that was on average 4 kHz lower than the tinnitus edge frequency affected the overall stimulus context of the whole experiment, and therefore shifted the response pattern at the tinnitus edge frequency even in blocks where control stimuli were not presented. It has been suggested that MMN is an indicator encompassing all dimensions of the stimuli presented, or combinations thereof, which could have different representations on the cortical surface depending on the paradigm features and would be advantageous in terms of survival in unexpected or improbable events [26–29]. While there is a plethora of

research indicating the importance of immediate context preceding a deviant stimulus in terms of the ERP waveform shape, not much is known about any differences in evoked response waveforms in contexts of longer time periods, e.g. minutes [26, 30–32]. Whilst it is speculative at this stage, we wonder whether the different experimental context here (a wider difference between frequencies) had a relatively larger effect on control subjects than tinnitus subjects, making their responses at the tinnitus frequency more 'tinnitus-like'. This could indicate that tinnitus-driven and context-driven effects could share a common mechanism. However, other potential reasons for the discrepant results in this study compared to the original study also exist.

### 4.4 Tinnitus subjects had higher scores on hyperacusis questionnaires

Hyperacusis causes normal environmental sounds to be uncomfortably loud [33]. Tinnitus and hyperacusis are often comorbid, but there are some distinctions in the auditory pathway changes related to each condition [34, 35]. There are difficulties with finding participants with tinnitus who do not also have some sound sensitivity; higher THI scores have been found to associate with co-occurrence of hyperacusis [36, 37]. The addition of hyperacusis has been shown to affect resting state EEG activity, compared to participants with only tinnitus [33, 37, 38]. The presence of hyperacusis has also been found to enhance the average sound-evoked activity both to frequencies that are affected by hearing loss and those that are intact, in subcortical and cortical structures, while reducing the responses to tinnitus frequency specifically when compared to a group that did not report having hyperacusis [36, 39–43]. Therefore, the findings in the current study may also be affected by this factor. The HQ scores were not related to the perceived loudness of stimuli in this study, however; the relationship between uncomfortable loudness level (ULL) and HQ scores has previously shown to be weak (e.g. [44, 45]) and the differences in stimulus sensation level between groups is likely to have been compensatory for the hyperacusis and therefore to allow the stimuli to be perceptually similar to control group. We specifically used a subjective method for stimulus intensity determination, to try and minimise the effect of hyperacusis on intensity MMN. However, it may be that it is not possible to fully account for the impact of hyperacusis in this way; in future studies, it may be important to further distinguish between hyperacusis and tinnitus and the combination of the two conditions, through careful subject group selection, and how brain responses to the current paradigm are affected by these [34].

## 5. Conclusion

The current study failed to replicate the Intensity Mismatch Asymmetry as a marker of tinnitus status [1]. The new findings could potentially be due to the overall frequency context of the paradigm affecting the responses to deviant stimuli, particularly in the control group, or other subject factors or technical aspects. Another potential reason for the discrepant findings could simply be the smaller sample size, however it is important to note that this study shows that the strength of IMA is at least smaller than it previously appeared, and in its present form far from the intended biomarker reliable at the individual subject level. An interesting next step would be investigating the effects of varying study contexts (e.g. large, small, or no difference between frequencies used in different blocks of the paradigm). This, and systematic exploration of other contributory factors, and other paradigm variants, may help to improve the diagnostic accuracy of prediction violation-based tinnitus biomarkers in future.

## Supporting information

**S1 Table. Tinnitus and tinnitus edge frequencies for each participant.**
(DOCX)

**S1 Checklist. STROBE statement—Checklist of items that should be included in reports of observational studies.**
(DOCX)

## Author Contributions

**Conceptualization:** William Sedley.

**Data curation:** Ekaterina A. Yukhnovich.

**Formal analysis:** Ekaterina A. Yukhnovich, William Sedley.

**Funding acquisition:** William Sedley.

**Methodology:** Kai Alter, William Sedley.

**Project administration:** Ekaterina A. Yukhnovich.

**Supervision:** Kai Alter, William Sedley.

**Visualization:** Ekaterina A. Yukhnovich.

**Writing – original draft:** Ekaterina A. Yukhnovich.

**Writing – review & editing:** Kai Alter, William Sedley.

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
