## [Decision Letter · Decision Letter 0]

7 May 2023

PONE-D-23-06918Nuances in intensity deviant asymmetric responses as a biomarker for tinnitusPLOS ONE

Dear Dr. Yukhnovich,

Thank you for submitting your manuscript to PLOS ONE. After careful consideration, we feel that it has merit but does not fully meet PLOS ONE’s publication criteria as it currently stands. Therefore, we invite you to submit a revised version of the manuscript that addresses the points raised during the review process.

We look forward to receiving your revised manuscript.

Kind regards,

Prashanth Prabhu

Academic Editor

PLOS ONE

“This work was supported by the Royal National Institute for Deaf People and the Masonic Charitable Foundation, United Kingdom. The authors declare no competing financial interests.”

“This work was supported by the Royal National Institute for Deaf People and the Masonic Charitable Foundation, United Kingdom (EAY, WS). The funders had no role in study design, data collection and analysis, decision to publish, or preparation of the manuscript.”

Reviewers' comments:

Reviewer's Responses to Questions

**Comments to the Author**

1. Is the manuscript technically sound, and do the data support the conclusions?

Reviewer #1: Yes

Reviewer #2: Yes

2. Has the statistical analysis been performed appropriately and rigorously? 

Reviewer #1: Yes

Reviewer #2: Yes

3. Have the authors made all data underlying the findings in their manuscript fully available?

Reviewer #1: Yes

Reviewer #2: Yes

4. Is the manuscript presented in an intelligible fashion and written in standard English?

Reviewer #1: Yes

Reviewer #2: Yes

5. Review Comments to the Author

Reviewer #1: Sampling size can be increased for future investigation purposes. Although the current study failed to replicate the Intensity mismatch asymmetry as a marker of tinnitus status but the new findings are still useful.

Reviewer #2: PONE-D-23-06918 Nuances in intensity deviant asymmetric responses as a biomarker for tinnitus

Justification of sample size.

Specify the age range of participant (both control and clinical group)

What is the starting level of intensity of band pass noise in psychophysical assessment?

How the psychophysical assessment performed in bilateral tinnitus.

Provide the details of tinnitus pitch, center frequency of tinnitus of each participant.

What is the duration of standard stimulus?

The experimental design closely followed the paradigm used (1), with the exception of using a 1 kHz tone as the control condition - consider to rephrase

The tones were presented isochronously to the ear(s) that the tinnitus participant indicated as the tinnitus ear(s), or the same ear(s) for their matched control (illustrate it with suitable example).

In Method - ‘with chronic tinnitus for over 6 months’ and in results According to the THI, tinnitus was causing a slight problem to four participants, mild problem to seven participants, a moderate problem to two participants and a severe problem to one participant. - Conflicting statement and THI representation (27.29 (16.69)*) in Table-2.

Tinnitus duration ranged from 4 to 60 years and in table-2 the duration in years 21 (19.16) - check it.

Mention the tinnitus frequency and edge frequency for each participant in a table..

There is a mismatch between Participant age, hearing loss pattern and tinnitus duration range. Do the participants in control groups are older adults,in audiogram it is observed that > 4 kHz above >25 >30 dB HL (Figure-1).

Specify the ocular channel used in recording the EEG data.

The mean dB difference between the loud and quiet stimuli was not significant between groups, however (Man U test, p =0.376) - Rephrase the sentence.

Possible reasons for the findings are missing under each heading of the discussion section.

6. PLOS authors have the option to publish the peer review history of their article (what does this mean?). If published, this will include your full peer review and any attached files.

Reviewer #1: **Yes: **Archana Gupta

Reviewer #2: **Yes: **Hemanth Narayan Shetty, Professor in Audiology, JSS ISH, Mysuru

---

## [Author Response · Author response to Decision Letter 0]

22 Jun 2023

Dear editors,

Thank you for reviewing our manuscript and providing constructive comments. I am very grateful for your time and suggestions. In this letter, I aim to address the issues brought up by the journal and the reviewers. 

1) “Funding information should not appear in the Acknowledgments section or other areas of your manuscript.”

I have removed the funding section from the main manuscript file (line 493). I also included a funding statement in the cover letter.

2) “Please review your reference list to ensure that it is complete and correct.”

I believe that my reference list is complete and does not include any retracted papers. 

Responses to reviewer 1:

1) “Sampling size can be increased for future investigation purposes.” 

Thank you for your comment! While the sample size could have been larger, we believe that it should have been sufficient to attempt replication of previous findings. I have added a statement about this to the manuscript (line 106): “The sample size was small largely because this study was seeking the kind of large and invariant effects indicated in the previous study”.

Responses to reviewer 2:

1) “Justification of sample size” 

While the sample size could have been larger, we believe that it should have been sufficient to attempt replication of previous findings. I have added a statement about this to the manuscript (line 106): “The sample size was small largely because this study was seeking the kind of large and invariant effects indicated in the previous study”.

2) “Specify the age range of participant (both control and clinical group)” 

I have now put this information in Table 1 (line 226; tinnitus group range 20-80 and control group range 22-70)

3) “What is the starting level of intensity of band pass noise in psychophysical assessment?”

I added this (line 134-136): “The intensity was based on an inverse Fourier transform of a Hanning spectrum noise, with peak amplitude at the centre frequency equal to 1. It was not a specific dB value, and it was always a relatively quiet stimulus which subjects needed to increase in intensity for the match.”

4) “How the psychophysical assessment performed in bilateral tinnitus.” 

I have added this on line 133: “In cases of bilateral tinnitus, the participants heard the sound in both ears, and could adjust the ear balance.”

5) “Provide the details of tinnitus pitch, center frequency of tinnitus of each participant.”

I have now put this in as a supplemental table.

6) “What is the duration of standard stimulus?” 

300 ms (line 171)

7) “The experimental design closely followed the paradigm used (1), with the exception of using a 1 kHz tone as the control condition - consider to rephrase”

I have now changed this to “The experimental design closely followed the paradigm used in the original study (1), with the additional inclusion of a 1 kHz tone as a control condition.” (line 167)

8) “The tones were presented isochronously to the ear(s) that the tinnitus participant indicated as the tinnitus ear(s), or the same ear(s) for their matched control (illustrate it with suitable example).”

I added “For example, if the participant with tinnitus only had tinnitus in their right ear, the matched control would also only hear the tones in their right ear.” (line 174).

9) “In Method - ‘with chronic tinnitus for over 6 months’ and in results according to the THI, tinnitus was causing a slight problem to four participants, mild problem to seven participants, a moderate problem to two participants and a severe problem to one participant. - Conflicting statement and THI representation (27.29 (16.69)*) in Table-2.”

This seems consistent to me; the largest frequency of participants scored between 18 and 36 (mild problem), which is reflected by the mean score being within this category also. I have included the ranges of category scores in the description (line 212): “According to the THI, tinnitus was causing a slight problem to four participants (<16 THI score), a mild problem to seven participants (18-36), a moderate problem to two participants (38-56) and a severe problem to one participant (58-76). Tinnitus duration ranged from 4 to 60 years. The duration was skewed to the left, so the mean (21) was lesser than could be expected in a normal distribution.”

10) “Tinnitus duration ranged from 4 to 60 years and in table-2 the duration in years 21 (19.16) - check it.”

This is also accurate based on my data. I have included an explanation for this (line 214): “the duration was skewed to the left, so the mean is lesser than could be expected in a normal distribution”.

11) Mention the tinnitus frequency and edge frequency for each participant in a table.

This is now in the supplemental table based on comment 5).

12) “There is a mismatch between Participant age, hearing loss pattern and tinnitus duration range. Do the participants in control groups are older adults, in audiogram it is observed that > 4 kHz above >25 >30 dB HL (Figure-1).”

The figure is representative of the PTA results I saw, with many participants starting to have some hearing loss from 6 kHz. The largest tinnitus duration occurred in one of my oldest participants, who indicated that they developed tinnitus as a teenager. The age and hearing loss means, and overall distributions were not significantly different between groups (line 233).

13) Specify the ocular channel used in recording the EEG data.

I have now included this (line 164): “No ocular channel was used as the standard EEG channels were sufficient for removing ocular artifacts”.

14) The mean dB difference between the loud and quiet stimuli was not significant between groups, however (Man U test, p =0.376) - Rephrase the sentence.

Changed this to “Nonetheless, the mean dB difference between the loud and quiet stimuli was not significant between the tinnitus and control groups (Man U test, p =0.376).” (Line 250).

15) Possible reasons for the findings are missing under each heading of the discussion section.

Thank you for the good suggestion. Upon further consideration, given that my discussion subsections are quite short, I am not sure that a summary at the beginning of each section would be needed, and I would prefer to avoid repetition.

Thank you to the journal for considering this paper and to the reviewers for their comments! I hope this response is satisfactory to any concerns raised. 

Kind regards,

Ekaterina Yukhnovich

---

## [Decision Letter · Decision Letter 1]

11 Jul 2023

Nuances in intensity deviant asymmetric responses as a biomarker for tinnitus

PONE-D-23-06918R1

Dear Dr. Yukhnovich,

We’re pleased to inform you that your manuscript has been judged scientifically suitable for publication and will be formally accepted for publication once it meets all outstanding technical requirements.

Kind regards,

Prashanth Prabhu

Academic Editor

PLOS ONE

Additional Editor Comments (optional):

Reviewers' comments:

Reviewer's Responses to Questions

**Comments to the Author**

1. If the authors have adequately addressed your comments raised in a previous round of review and you feel that this manuscript is now acceptable for publication, you may indicate that here to bypass the “Comments to the Author” section, enter your conflict of interest statement in the “Confidential to Editor” section, and submit your "Accept" recommendation.

Reviewer #1: All comments have been addressed

2. Is the manuscript technically sound, and do the data support the conclusions?

Reviewer #1: Yes

3. Has the statistical analysis been performed appropriately and rigorously? 

Reviewer #1: I Don't Know

4. Have the authors made all data underlying the findings in their manuscript fully available?

Reviewer #1: Yes

5. Is the manuscript presented in an intelligible fashion and written in standard English?

Reviewer #1: Yes

6. Review Comments to the Author

Reviewer #1: (No Response)

7. PLOS authors have the option to publish the peer review history of their article (what does this mean?). If published, this will include your full peer review and any attached files.

Reviewer #1: No

---

## [Editor Report · Acceptance letter]

27 Jul 2023

PONE-D-23-06918R1 

Nuances in intensity deviant asymmetric responses as a biomarker for tinnitus 

Dear Dr. Yukhnovich:

I'm pleased to inform you that your manuscript has been deemed suitable for publication in PLOS ONE. Congratulations! Your manuscript is now with our production department. 

Kind regards, 

on behalf of

Dr. Prashanth Prabhu 

Academic Editor

PLOS ONE